# Comprehensive Phenotypic Characterization of Diverse Drug-Type *Cannabis* Varieties from the Canadian Legal Market

**DOI:** 10.3390/plants12213756

**Published:** 2023-11-02

**Authors:** Éliana Lapierre, Maxime de Ronne, Rosemarie Boulanger, Davoud Torkamaneh

**Affiliations:** 1Département de Phytologie, Université Laval, Québec, QC G1V 0A6, Canada; eliana.lapierre.1@ulaval.ca (É.L.); maxime.de-ronne.1@ulaval.ca (M.d.R.); rosemarie.boulanger.1@ulaval.ca (R.B.); 2Institut de Biologie Intégrative et des Systèmes (IBIS), Université Laval, Québec, QC G1V 0A6, Canada; 3Centre de Recherche et d’Innovation sur les Végétaux (CRIV), Université Laval, Québec, QC G1V 0A6, Canada; 4Institut Intelligence et Données (IID), Université Laval, Québec, QC G1V 0A6, Canada

**Keywords:** cannabis, phenotype, breeding, correlations, selection

## Abstract

*Cannabis* (*Cannabis sativa* L.) stands as a historically significant and culturally important plant, embodying economic, social, and medicinal relevance for human societies. However, years of prohibition and stigmatization have hindered the cannabis research community, which is hugely undersized and suffers from a scarcity of understanding of cannabis genetics and how key traits are expressed or inherited. In this study, we conducted a comprehensive phenotypic characterization of 176 drug-type cannabis accessions, representative of Canada’s legal market. We assessed germination methods, evaluated various traits including agronomic, morphological, and cannabinoid profiles, and uncovered significant variation within this population. Notably, the yield displayed a negative correlation with maturity-related traits but a positive correlation with the fresh biomass. Additionally, the potential THC content showed a positive correlation with maturity-related traits but a negative correlation with the yield. Significant differences were observed between the plants derived from regular female seeds and feminized seeds, as well as between the plants derived from cuttings and seeds for different traits. This study advances our understanding of cannabis cultivation, offering insights into germination practices, agronomic traits, morphological characteristics, and biochemical diversity. These findings establish a foundation for precise breeding and cultivar development, enhancing cannabis’s potential in the legal market.

## 1. Introduction

Cannabis (*Cannabis sativa* L.) was one of the first domesticated plants [1], with a rich and dynamic history closely linked to the economic, social, and cultural progress of human societies [2]. Cannabis, belonging to the Cannabaceae family, is an annual herbaceous plant [3,4,5,6]. Generally, it is dioecious (2n = 20), characterized by nine autosomes and one pair of sex chromosomes. However, the process of sex determination in cannabis is rather intricate; most drug-type cannabis plants are dioecious, exclusively producing either male (XY) or female (XX) flowers [7,8,9]. On the other hand, some plants, primarily hemp, exhibit monoecious characteristics, with both male and female flowers growing on the same plant [10]. Under specific conditions, dioecious plants may revert to a monoecious-like state, known as hermaphroditism [7,11,12]. In cannabis, sex holds a great importance: unpollinated female flowers produce and store considerably more cannabinoids than male flowers [7,8,13].

Cannabis was a versatile resource, providing fibers for ropes and nets, food, and oil from its seeds for millennia, while the only direct evidence supporting the medicinal use of cannabis dates back to ~400 A.D., based on the discovery of delta-6-tetrahydrocannabinol (D^6^-THC) in ashes [14]. In Canada, similar to the U.S.A. and the E.U., cannabis is divided and regulated based on the level of the psychoactive cannabinoid D^9^THC produced in the plant (plants with below 0.3% are regulated as hemp and 0.3% or greater as drug-type) [15]. While hemp has been legal for commercial cultivation in Canada since 1998, drug-type cannabis was only legalized for commercial production in 2014 for medicinal and 2018 for recreational use [16]. The global legal cannabis market is experiencing exponential growth, projected to reach a staggering $102 billion by 2028, up from the current value of $51 billion [17], making it one of the most economically significant crops worldwide. However, despite this rapid commercial expansion, the biology of cannabis remains shrouded in mystery, largely due to its prolonged history of prohibition. The scarcity of empirically based best practices and optimized protocols across the entire cannabis industry is undeniable. From breeding and genetics to cultivation, processing, and postharvest practices, there exists a dearth of comprehensive knowledge and understanding. The legacy of prohibition has hindered scientific exploration and research, leaving us with a vast realm of untapped potential and untamed possibilities.

Cannabis is known to produce over 545 potentially bioactive secondary metabolites, including more than 120 cannabinoids, various flavonoids, and a plethora of terpenes [18,19,20]. While clinical studies are typically conducted with purified cannabinoids (D^9^THC and/or cannabidiol (CBD)), or with products standardized with only these two compounds (i.e., marker compounds), many patients and recreational users (more than 80%) consume raw plant products or whole plant extracts and anecdotal reports have long claimed that the effects vary based on the broader chemical profile [21]. What makes whole plant extracts more challenging is that they are not just chemically complex, but also chemically variable, such that two products with identical levels of marker compounds can be chemically quite different and produce dramatically different medical outcomes [22]. This variation is due to a combination of the genetic background of the plant, as well as the environment in which it is grown, referred to as genotype by environment (GxE) interactions [23,24]. To achieve consistently reproducible extracts, it is imperative that they originate from genetically stable plants, such as inbred lines, cultivated under highly uniform conditions. Indeed, utilizing stabilized cultivars obtained through the inbreeding process, such as cyclic improvement [25], represent a unique opportunity for cannabis growers. This approach not only enables the production of uniform and consistent cannabis products but also allows for the exploration of novel combinations of cannabis genetics. By leveraging stabilized cultivars, growers can develop varieties with enhanced disease resistance, novel flavors, and aromas, as well as tailored cannabinoid levels to meet specific market demands. Additionally, this method empowers growers to optimize total yields, further enhancing the economic potential of cannabis cultivation.

In the realm of agriculture, cultivar development is advancing into a new era through the integration of modern breeding techniques, genomic tools, and genome editing [26]. The first and main step taken in most plant breeding platforms is inbreeding to fix desired traits. This is a challenge in cannabis, as cannabis plants are predominantly dioecious and highly heterozygous, exhibiting substantial variation within its population and final product [27]. Although cannabis is an outcrossing species with exceptionally high levels of within-population variability, clonal propagation methods are relatively easy and are used to mass-produce uniform populations [12]. As a result, clonal propagation methods have become the main approach for large-scale production. This method involves taking cuttings from mother plants with desirable characteristics to multiply and produce genetically identical offspring (clones). However, traditional clonal propagation poses challenges, such as occupying significant growing space, often more than 15%, limiting the mass production capacity, and making growers vulnerable to pests and pathogens that transmit from mother plants to cuttings [28]. Moreover, in a recent study, significant genetic diversity within an individual cannabis mother plant was discovered, demonstrating the ubiquity of somatic mutations in cannabis plants [29]. This finding aligned with anecdotal reports from cannabis producers indicating that clones deteriorated over time. Micropropagation, a promising alternative, offers a controlled and sterile environment using tissue culture techniques to produce genetically uniform clones free from pests and diseases. It allows for the preservation of valuable genetic material, facilitates genetic modification for improved traits, and offers disease-free plant production [11]. Yet, micropropagation also requires the addressing of challenges such as maintaining a sterile environment and potential somaclonal variations leading to mutations [30].

The development of cannabis inbred lines is a pivotal endeavor geared towards achieving genetic stability, predictability, and superior traits in the offspring. A stellar example of the potential of inbreeding and hybrid vigor lies in maize, where hybrids resulting from crossing two inbred lines exhibited significant heterosis, surpassing their counterparts in productivity and uniformity [31]. This groundbreaking practice led to a significant surge in maize yields over time, with nearly half of this progress credited to improved genetics, particularly the shift from open-pollinated cultivars to F_1_ hybrids derived from inbred lines [32]. A fundamental prerequisite for unleashing the potential of plant breeding is genetic variation, acting as a vital catalyst in the utilization of germplasm, which forms the genetic foundation for developing new and improved plant varieties [33]. As such, the initial and paramount step in this journey lies in the phenotyping of plant materials, a process that stands as a cornerstone of plant breeding. While the realm of genetics has witnessed remarkable advancements and the integration of cutting-edge molecular technologies in crop research, the art of crop breeding continues to rely extensively on the expression of agronomically important traits [34]. These traits not only govern the selection process, but they also dictate the defining features of commercial products. Phenotyping, with its deep-rooted emphasis on identifying and characterizing these vital traits, remains an indispensable tool in the hands of plant breeders. It provides valuable insights into the intricate tapestry of genetic variation that lays the groundwork for the cultivation of superior crops and the emergence of robust agricultural systems [35]. By unraveling the phenotypic landscape, plant breeders navigate the intricacies of selecting and honing traits that confer resilience, productivity, and quality, all essential attributes for developing new and improved varieties.

Given the multifarious factors that impact cannabis breeding and the necessity to discern observable distinctions within this species, our objective was to characterize a diverse panel of 210 cannabis accessions, which represents the Canadian legal market. This characterization took place under highly controlled conditions, enabling the measurement of both phenotypic traits (e.g., plant height, days to maturity, yield, etc.) and biochemical traits (e.g., cannabinoids profiles). This comprehensive endeavor lays the groundwork for a drug-type cannabis breeding program, facilitating the development of novel and improved cannabis varieties with desired characteristics.

## 2. Results

### 2.1. Optimizing Seed Germination in Cannabis: No Treatment Is the Best Treatment

Here, our focus was to find the optimal sowing method for cannabis seeds, achieved through the exhaustive exploration of 13 distinct treatments (Appendix A). Upon subjecting the data to analysis of variance (ANOVA), intriguingly, we observed no statistically significant differences in the germination rates among the treatments (*p* = 0.08). However, despite not being significantly different, the direct sowing method (SD) displayed the numerically highest germination rate (Figure 1 and Appendix A). Therefore, this led us to opt for the direct sowing method for the phenotypic characterization trials, as it showed a better and more uniform germination rate compared to other methods that involved the pretreatments of seeds.

### 2.2. Exploring Diversity: Significant Variation in Agronomic Traits

We conducted a comprehensive characterization of six pivotal agronomic traits in a population of 176 cannabis accessions, representing the cannabis drug-type legal market (Appendix A). The key agronomic traits for the breeding program, including days to sexual maturity (dtms), days to maturity from flower initiation (dtmf), and the yield, exhibited significant variation (Figure 2). For instance, the days to maturity from sowing or cutting initiation (dtmp) ranged from 70 to 133 days, while the dtmf displayed a variation spanning 38 to 123 days. In a similar vein, the yield demonstrated a tapestry of remarkable diversity, stretching from a minimum of 6.02 g to an impressive pinnacle of 177.03 g (Figure 2).

In order to thoroughly assess the relationships among diverse agronomic traits, an analysis of the correlation was conducted. As can be seen in Figure 2, significant positive correlations were found for key selection traits such as days to first flower (dtf) and dtsm (r = 0.82), indicating a compelling interconnectedness. Conversely, for the yield, a distinct and significant negative correlation was found with dtsm (r = −0.55) and dtmp (r = −0.45), highlighting an intriguing relationship where certain traits influence the yield. In addition, the fresh biomass (fb) exhibited a highly significant and positive correlation with the yield (r = 0.81). The revelation of these correlations enriches our understanding of the complex web of interactions within the realm of cannabis agronomy, paving the way for more informed decision-making and strategic interventions in our selection program.

### 2.3. Exploring Diversity: Complex Interactions among Morphological Traits

In this study, we performed an in-depth characterization for seven morphological traits. Our findings revealed that the plant height at harvest (hh), growth index of the canopy diameter (gicd), and growth index of the internode length (giinl) data followed a normal distribution pattern, while the other traits displayed distributions that deviated from normality (Figure 3 and Appendix A). Significant differences were observed among the morphological traits of interest for selection. For instance, the growth index of the height (gih) ranged from 1 inch to 36.50, the gicd varied from 0 inches to 22 inches, and the number of nodes at harvest (nodeNH) ranged from a minimum of 18 to a maximum of 44 nodes. As depicted in Figure 3, notable and consistent positive correlations were identified across all morphological traits. Particularly noteworthy correlations emerged, such as those between the giinl and gih (r = 0.82), gih and nodeNH (r = 0.77), and giinl and nodeNH (r = 0.67) and stem diameter at harvest (sdah) and growth index of the stem diameter (gisd) (r = 0.60) demonstrating a significant interplay among these various morphological traits. Across the 176 accessions studied, we also observed a striking morphological diversity in various aspects, such as the inflorescence morphology and leaf characteristics. The wide-ranging array of inflorescence structures shown in Appendix A illustrating the significant variability that exists within this cannabis population.

### 2.4. Unlocking Cannabis Chemistry: Comprehensive Biochemical Analysis

Following the procedures of the drying and trimming of the harvested flowers, a series of representative samples were prepared for the biochemical analysis. This comprehensive analysis was focused on the quantification of 11 distinct cannabinoids: tetrahydrocannabinolic acid (THCA), delta-9-Tetrahydrocannabinol (D^9^THC), cannabidiolic acid (CBDA), cannabidiol (CBD), cannabigerolic acid (CBGA), cannabigerol (CBG), cannabichromene (CBC), cannabinol (CBN), tetrahydrocannabivarin (THCV), cannabidivarin (CBDV), and delta-8-Tetrahydrocannabinol (D^8^THC), in addition to potential THC (THCP) and CBD (CBDP) content. To ensure the robustness of the analytical method, a validation trial was conducted. The outcomes of this intricate validation test have yielded remarkably promising results. A compelling narrative of minimal variation, not statistically significant, has been etched, with the standard deviation (SD) encompassing a range of 3% to 8% across the different metabolites within the replicates (Appendix A). We were able to obtain concentrations exceeding the minimal detectable quantity for seven cannabinoids: THCA, D^9^THC, CBDA, CBD, CBGA, CBG, and CBC. For CBN, THCV, CBDV, and D^8^THC, a qualitative assessment was provided with presence and absence data. As a result, these four cannabinoids were not included in our statistical analysis.

Following a rigorous validation process, the entire population, comprising 176 samples, underwent a comprehensive analysis of their cannabinoid profiles. Remarkably, significant variations in the major cannabinoids, specifically THC and CBD, were observed. The THCP content oscillated within a striking range of 0.29% to 32.62%, while CBDP exhibited a span of 0.01% to 18.1%. As anticipated, strong positive correlations were observed between THCP-THCA, CBDP-CBDA, and CBD-CBDP (r = 0.97, r = 1, and r = 0.96, respectively), as depicted in Figure 4. Similarly, the interplay between CBC-CBDA bore a substantial positive correlation of 0.88. On the contrary, moderate negative correlations were also detected, underscoring the dynamics of the cannabinoid landscape. Specifically, CBDP-THCA (r = −0.61), THCP-CBDA (r = −0.63), THCP-CBD (r = −0.65), CBC-THCP (r = −0.59), and THCP-CBDP (r = −0.63) exhibited discernible negative linear relationships. These findings deepen our understanding of cannabinoid interactions and pave the way for advanced research in cannabis chemistry.

### 2.5. Influence of Agronomic and Morphological Characteristics on Major Cannabinoids

Positive correlations were identified in this study, with a correlation coefficient (r) of 0.27 between THCP and the giinl (Appendix A). Additionally, an intriguing positive correlation (r = 0.64) emerged between the hh and both the yield and fb, mirroring a similar trend for the cdh, with coefficients of 0.42 and 0.53, respectively. Notably, positive correlations were found between the dtsm, dtf, dtmp, and the THCP. Conversely, a negative correlation of −0.24 was observed between the yield and the THCP. Notably, no significant correlations were detected between the agronomic and morphological characteristics and the CBDP (Appendix A).

### 2.6. Impact of the Origin of the Cannabis on Agronomic, Morphological, and Biochemical Traits

#### 2.6.1. Regular Female vs. Feminized Seeds

We found statistically significant differences between the plants derived from regular female (reg) seeds and feminized (fem) seeds, particularly in the dtmp, THCA, and THCP (*p*-values 2.20 × 10^−16^, 1.72 × 10^−7^ and 1.34 × 10^−7^, respectively) (Appendix A). The dtf and gin also exhibited significant differences, albeit with slightly higher *p*-values (2.20 × 10^−16^ and 6.45 × 10^−5^, respectively) between the female and feminized plants. Intriguingly, on average, the plants originating from regular seeds exhibited a higher THCP content (20.07%) compared to those from feminized seeds (15.94%). A similar pattern is evident in the THCA content, reflecting a notable 4.74% rise in the plants sourced from regular seeds as opposed to feminized seeds. Regarding the agronomic trait dtmp, the plants from regular seeds took longer to mature (109 days) than those from feminized seeds (104 days). Finally, the principal component analysis across the 13 traits over our 176 individuals, when organizing the data by their sex origin, showed no distinct clustering within the population, suggesting a nuanced interplay between the two seed types (Figure 5A).

#### 2.6.2. Cuttings vs. Seeds

The PERMANOVA analysis revealed notable distinctions between the type of germplasm sources, cuttings, and seeds. These differences were particularly pronounced and statistically significant for certain traits, specifically the nodeNh (W = 0.94) and dtsm (W = 0.88), accompanied with a very low *p*-value of 1.65 × 10^−10^. This unequivocal result underscores the substantial dissimilarity between the two designated source types (Appendix A). Importantly, the origin of the plants, whether derived from cuttings or seeds, exerts an influence over other traits. CBDA, CBD, gih, CBC, and CBDP, all stand out, exhibiting remarkable statistical significance with *p*-values falling below the threshold of 0.05. The plants cultivated from seeds demonstrated notably greater height growth, with growth indices of 17.86 inches compared to 7.91 inches for plants grown from cuttings. This trend is consistent for the nodeNH as well, with seed-grown plants averaging 34 nodes while cutting-grown plants averaged 24 nodes. Additionally, seed-grown plants took an average of 27 days longer to achieve sexual maturity (dtsm), although no significant distinction between the two source types was observed regarding maturity attainment from the initiation of flowering. On a biochemical level, the plants from cuttings exhibited markedly higher levels of CBDA, CBD, CBC, and CBD potential compared to their seed counterparts. Furthermore, the principal component analysis demonstrated a distinct clustering pattern, effectively categorizing the plants into two distinct groups: those derived from cuttings and those from seeds (Figure 5B). This collective insight emphasizes the far-reaching impact of the germplasm source on diverse traits within the cannabis accessions.

## 3. Discussion

Since its legalization, the cannabis industry has made a remarkable contribution to the gross domestic product (GDP), totaling 43.5 billion dollars, and sustaining over 151,000 jobs in Canada [36]. As the cannabis production modernizes and the demand for reliable and scientifically based cultivation rises, the necessity for informed genetic improvement to create pure and stabilized cultivars for therapeutic and commercial purposes becomes paramount [25]. As for any breeding program, target traits can be harnessed from the inherent diversity found within the species, employing well-phenotyped and characterized accessions as valuable resources. However, this is challenging in cannabis as it is predominantly dioecious and highly heterozygous, exhibiting substantial variation within populations [5,11,27]. Yet another obstacle lies in the availability of cannabis seeds, with the absence of a public gene bank for breeders and the considerable expenses linked to seed procurement [15]. Compounded by the restricted quantity of seeds (ranging from 1 to 10 seeds per accession), ensuring optimal germination processes became imperative for our endeavors. Despite its significance, a scientifically established method for cannabis seed germination has yet to be developed [37]. Hence, we undertook the comprehensive testing of every treatment available in both the legacy market and analogous species. Interestingly, the direct sowing of cannabis seeds demonstrated the highest level of uniformity and a superior germination rate. This pattern differed from the available studies where seed treatments generally yielded higher germination rates than the control group [37,38,39,40,41,42,43]. Notably, none of the aforementioned studies provided a comparison involving direct seeding in a growth substrate. It is also important to mention that most studies focused on sterilization treatment methods within an in vitro culture context instead of a greenhouse context [37,41,43].

In this study, we have conducted an extensive phenotypic and chemotypic characterization on an unprecedented scale, examining 176 cannabis accessions, representative of the Canadian drug-type cannabis legal market, revealing substantial diversity and correlations between the 176 accessions. As demonstrated, a strong correlation between the fresh biomass and the yield was found, meaning that plants with a higher fresh biomass tend to also exhibit a higher yield of dried flowers, consistent with previous studies [44]. We also found a significant positive correlation between the days to maturity and days to sexual maturity indicating that the longer a plant takes to exhibit signs of sexual maturity—characterized in the industry by the emergence of floral structures during the vegetative phase along the main stem nodes—the later it will reach harvest maturity. Similar to Naim-Feil et al. (2021), our findings revealed a significant negative correlation between some of agronomical traits (dtsm and dtmp) and yield. In their paper, a non-significant correlation of −0.12 was found between the days to maturity and the dried bud weight, suggesting that plants that take longer to reach sexual maturity and harvest maturity tend to yield less when it comes to the weight of the harvested dried flowers. These findings lay the groundwork for future breeding endeavors to consider these intricate relationships.

The characterization of five morphological traits further unveiled significant variations among cannabis plants. A strong positive correlation between the growth index of the internode length and growth index of the height was documented, indicating that as the internode lengths increase, the overall plant height also tends to increase [45,46]. Additionally, the positive correlation between the growth index of the internode length and the number of nodes at harvest suggests that plants with longer internodes might also exhibit a higher node count, potentially indicating a combined effect of internode elongation and increased branching on the overall plant structure. The overarching architecture of the cannabis plant holds significant importance, both influencing the yield and serving as a constraint in indoor production systems. The insights from this study illuminate the interplay of diverse traits and offer valuable insights for potential exploitation.

A comprehensive analysis of 11 distinct cannabinoids and the potential THC and CBD content in our population revealed substantial variations in cannabinoid profile [47]. Significant positive and negative correlations among these compounds provided insights into their interactions and complexities and is consistent with previous studies [48,49,50,51]. The strong positive correlations observed between THCP-THCA, CBDP-CBDA, and CBD-CBDP suggest a consistent and coordinated relationship between these cannabinoid pairs as expected. With THCA and CBDA being precursors of the potential THC and CBD, this strong correlation was anticipated and well-documented in the literature [18,21,44,46,52,53]. The negative correlations observed between CBDP-THCA, THCP-CBDA, THCP-CBDP, THCP-CBD, CBC-THCP were also expected as THC and CBD share the same precursor, CBGA [46,51,54]. The influence of morphological and agronomic traits on major cannabinoids (THCP and CBDP) play a key role in cannabis breeding programs. Positive correlations were found between THCP and certain traits, shedding light on potential links between plant morphology and cannabinoid production. In contrast to previous studies [46,51] we found a slight correlation (r = 0.27) between THCP and the growth index of the internode length, which suggests that there is a slight tendency for plants with higher THCP to also have slightly longer internodes. These findings underline the importance of plant size and growth characteristics in optimizing cannabis cultivation for higher productivity.

Similarly, we found positive relationships between certain agronomic traits, such as days to sexual maturity (dtsm), days to first flowers (dtf), and days to maturity from sowing/cutting (dtmp), and the THCP. These correlations suggest that there might be some influence of the plant’s growth timeline on its THCP, although the relationships are not extremely strong and could be influenced by other factors as well, as corroborated by Stack et al. (2021).In contrast to the findings of Naim et al. (2023), who suggested that breeding for high-yielding plants might not be confined to a specific chemotypic composition, our study reveals an intriguing dynamic between the cannabinoid content and yield in cannabis. Specifically, we observed a significant negative correlation between THCP and the yield. Our findings emphasize the necessity for nuanced breeding strategies that account for the intricate relationships between cannabinoid profiles and yield.

Feminized seeds and regular seeds are two types of cannabis seeds that differ primarily in their genetic makeup and reproductive characteristics [8]. Regular seeds are of high importance for breeders and growers to develop new varieties. Regular seeds exhibit a roughly 25% likelihood of producing male plants, which can pose challenges as non-fertilized female plants are preferred due to their higher cannabinoid production. In contrast, cannabis’s flexible sexuality has led to the development of an intelligent approach to masculinize female plants, using foliar sprays of specific plant growth regulators or inhibitors (e.g., silver thiosulfate) [8], to produce female seeds, so-called feminized seeds. Significant differences between the plants derived from regular female seeds and feminized seeds were found in this study. The comparison yielded noteworthy and statistically significant differences, particularly evident in key traits such as days to maturity from sowing/cutting (dtmp), THCA content, and THCP. All means for these traits were notably higher in the case of the plants from regular female seeds. This pattern suggests that plants from regular female seeds tended to exhibit extended maturity periods from sowing/cutting, higher concentrations of THCA, and greater THCP compared to their feminized seed counterparts. However, no differences were found between the two sex types for traits such as the yield, CBDP, fresh biomass, gih, etc. These findings highlight a new substantial impact of the source type on these specific traits and underscore the potential implications for cannabis cultivation and breeding strategies.

Moreover, we found significant differences between the use of cuttings and seeds as sources of germplasm. Notably, traits such as the nodeNH and dtsm displayed considerable differences. These differences suggest potential advantages in terms of plant size and developmental timing for seed-derived plants. Interestingly, no significant distinctions were observed between the two source types in terms of days to maturity from sowing/cutting, days to maturity from flower initiation, the fresh biomass, yield, and THCP. These findings indicate that while certain morphological and agronomical traits may exhibit variability based on the source of the germplasm, key traits for the industry and breeding such as the yield and days to maturity remain consistent between plants derived from seeds and cuttings. However, the observed distinction between the plants derived from cuttings and those from seeds had a notable impact on the cannabinoid profile. Specifically, the concentrations of CBDA, CBD, CBC, and CBDP exhibited statistically significant differences, with higher mean concentrations observed in the plants derived from cuttings. The significant differences in the cannabinoid concentrations suggest that the genetic makeup of the plants, influenced by the source of the germplasm, plays a crucial role in determining the resulting cannabinoid content. Nevertheless, it is crucial to emphasize that additional research is essential to validate these findings, ideally through the use of isolines for more conclusive results. This tendency was also observed in another study [55], where the authors found higher CBD and THC content for vegetative cutting plants than from seed derived plants. While it is important to note that there is a possibility that the observed cannabinoid profile differences in the cutting-derived plants might be influenced by the specific accessions chosen for the trial, further investigation is needed to determine any potential role of the genotype in contributing to these differences.

As cannabis cultivation becomes more scientific and demand-driven, strategic genetic enhancement becomes crucial. Our study aimed to establish foundational populations with quantified traits to guide breeding efforts. Overall, this study contributes to the intricate science of cannabis cultivation, offering a roadmap for informed breeding strategies and recognizing the nuanced interplay between genetics, growth, and chemical composition. Further research is warranted to refine these findings and advance cannabis cultivation practices.

## 4. Materials and Methods

### 4.1. Germination Test

To optimize the germination process of cannabis seeds and minimize external influences, this study focused on identifying effective pre-treatment methods. Extensive research was conducted through a literature review and consultation with industry professionals to gather insights on germination techniques [38,39,40,41,42,43,56]. From the available information, seven pre-treatment methods were carefully selected, comprising five chemical and two physical approaches, (see Appendix A for detailed descriptions) representing a total of 13 treatments: soaking in water for 18 or 24 h (W-18, W-24), soaking in 1 and 3% hydrogen peroxide (HP-1, HP-3), soaking in gibberellic acid solution 500 and 800 mg/L (GA-500, GA-800), soaking in 1% bleach solution for 2 and 24 h (NaCl-2, NaCl-24), soaking in potassium nitrate 1.5 and 3%, (KN-1.5, KN-3), knife scarification and soaking in water (K + W), sandpaper scarification and soaking in water (S + W), and direct sowing (SD).

Considering the challenges associated with procuring a sufficient quantity of homogenous drug-type cannabis seeds, an alternative approach was adopted. The germination tests were performed using seeds from an inbred hemp-typed cannabis cv. Vega. Vega is a grain hemp cultivar characterized by its short plant stature, exceptional lodging tolerance, uniform height, and flowering, and it typically exhibits a very low male count. The seeds from this cultivar contain approximately 0.2% THC and 1.6% CBD [57]. These seeds were produced in 2022 and obtained from Céréla Inc. in Quebec, QC, Canada. To ensure the reliability of our experimental setup, we subjected each pre-treatment method to three replicates, each consisting of 10 seeds, resulting in a total of 30 seeds for each treatment. The experimental setup took place in the high-performance greenhouse complex at Université Laval, offering precise control over environmental conditions to provide optimal growing conditions. The greenhouse maintained an 18 h light and 6 h dark cycle, with temperatures set at a constant 26 °C, and a relative humidity of approximately 50%, creating an ideal atmosphere for seed germination. To execute the pre-treatment methods, the seeds were treated with the respective solutions and subsequently germinated in sterile Petri dishes (Fisherbrand™ Petri Dishes with Clear Lid, Thermo Fisher Scientific, Waltham, MA, USA). The pre-treatments involved soaking the seeds in 15 mL of solution for different durations, depending on the specific treatment. After soaking, the seeds were rinsed with distilled water and then germinated for three days in Petri dishes with Whatman filter paper moistened with distilled water. Throughout the entire germination period, the Petri dishes were kept in darkness, and the germination rate was recorded daily for each treatment.

### 4.2. Plant Materials

Under our cannabis research license (LIC-QX0ZJC7SIP-2021) and in full compliance with Health Canada’s regulations, the seeds and cuttings used in this study were legally procured and imported from different sources. In this study, we assembled a population of 210 cannabis accessions, representing the drug-type cannabis legal market. Out of the 210 accessions, 62 accessions were obtained as 10-day-old cuttings from our research partner, Fuga Group Inc., Stoneham-et-Tewkesbury, QC, Canada, as we encountered challenges in finding a reliable seed source. For the remaining accessions, both feminized and regular seeds were cultivated. In the case of regular seeds, multiple seeds were germinated, and a PCR-based sex-determination test (see full description in the following Section 4.4.2 was conducted to selectively retain only the female plants. Due to various factors such as seed germination failures and the absence of female plants in certain accessions, the full phenotyping was carried out on 176 accessions (Appendix A).

### 4.3. Growing Conditions

The cannabis plants were cultivated in two separate 50 m^2^ fully blacked-out compartments within high-performance greenhouses at Université Laval. One compartment was dedicated to the vegetative stage, while the other was used for the flowering stage (Appendix A). The cultivation took place at three different time points, spanning from July 2022 to May 2023. Throughout the growth period, the plants were illuminated by high-performance FloraQueen LED lights (Futur-Vert Inc., Prévost, QC, Canada) positioned 2 feet above the plants. The light intensity was adjusted to maintain a maximum intensity of approximately 1000 μmol s^−1^ with an average light intensity of 550 μmol s^−1^. For the beginning of the vegetative stage, which includes germination, the plants were placed under HPS lights. These lights provided a maximum intensity of approximately 600 watts in the vegetative compartment (Appendix A). During flowering, the light intensity was measured daily using HOBO USB Micro Station Data Logger (H21-USB) and Solar Radiation (Silicon Pyranometer) (S-LIB-M003) and Photosynthetic Light (PAR) (S-LIA-M003) sensors installed in the greenhouse. During the vegetative stage, the temperature was set at 26 °C during the day and 22 °C during the night, with a relative humidity of 60% during the day and 55% during the night. In the flowering stage, the temperature was adjusted to 25 °C during the day and 21 °C during the night, with a relative humidity of 72% during the day and 70% during the night (Appendix A).

A conventional cannabis cultivation method was employed in this study. Briefly, at first, the seeds were germinated in multicellular trays using PROMIX-BX soil mix (Promix). Subsequently, the seedlings were transplanted into 4-inch, 1-gallon, and 5-gallon pots (Gérard Bourbeau & Fils, Québec, QC, Canada), utilizing organic Lambert EPM soil mix (Lambert) for the remainder of their growth (Appendix A). The plants underwent a vegetative phase for 5–7 weeks, starting from the sowing date or the reception of cuttings, depending on their height. Afterward, the plants were transitioned to the flowering phase under a 12 h photoperiod. The duration of the flowering phase ranged from 8 to 10 weeks, depending on the maturity progression.

Nutritional requirements were closely monitored and adjusted accordingly throughout the cultivation period, following a fertilization program designed by Marie-Odile Belley (a cannabis agronomist at Plant Products Inc., Québec, QC, Canada). The plants received daily fertigation with a nutrient solution consisting of various components, including PP-Optimum (12-2-14), PP-MJ Boost (15-30-15), chelated micronutrients from PP, potassium sulfate, ACTIV (0-0-5), and EZGro Armour (0-0-15) (Plant Products Inc., Quebec, QC, Canada). Soil pH and electrical conductivity (EC) were regularly monitored on a weekly basis, utilizing soil analyses performed by Plant Products Inc., Quebec, QC, Canada. The target soil pH range was maintained between 6 and 6.5, while the optimal EC levels were adjusted based on the growth stage (see details in the Appendix A). To provide support and prevent plant breakage, 42” tall metal cages were installed in the pots at the onset of the first flowering week. No physical modifications or alterations (such as pruning and topping) were made to the plants to avoid interfering with the expression of their natural phenotype. However, dead, damaged, or yellowing leaves, as well as those obstructing the inflorescences, were selectively removed at various stages of cultivation. Pest management strategies involved the introduction of predatory insects (*Amblyseius swirskii* ((by Anatis and Biobest)) and the application of Zerotol (BioSafe Systems) to prevent botrytis infection during the initial experiment.

### 4.4. DNA Extraction and Sex Determination

#### 4.4.1. Sample Collection and Preparation

For each accession, approximately 50 mg of young leaf tissues were collected for DNA extraction. Collected leaf tissues were dried for 4 days using a desiccating agent (Drierite; Xenia, OH, USA) and then ground with metallic beads in a RETSCH MM 400 mixer mill (Fisher Scientific, Waltham, MA, USA). DNA extraction was performed using the CTAB-chloroform protocol. In brief, the powdered tissue was treated with a CTAB buffer solution, followed by phenol-chloroform extraction. The resulting DNA pellet was washed with ethanol and re-suspended in water. DNA quantification was performed with a Qubit fluorometer using the dsDNA HS assay kit (Thermo Fisher Scientific, Waltham, MA, USA) and subsequently adjusted to 10 ng/µL for each sample.

#### 4.4.2. Polymerase Chain Reaction (PCR) for Sex Determination

Sex of cannabis individuals was predicted at the seedling stage using a PCR-based assay described by [58,59] with the following oligos: SCAR119_F:5′-TCAAACAACAACAAACCG-3′ and SCAR119_R: 5′-GAGGCCGATAATTGACTG-3′. DNA fragment analysis was performed on 1.5% agarose gel electrophoresed for 30 min (10 V/cm) in TAE buffer (0.4 M Tris acetate pH 8.3, 0.01 M EDTA). The gel was stained with SYBR safe DNA gel (Invitrogen, Waltham, MA, USA) and revealed with a gel imaging fluorescence system. Male cannabis individuals were identified by the occurrence of a 119 bp DNA fragment.

### 4.5. Phenotype Characterization

#### 4.5.1. Agronomic and Morphological Traits

Data on various agronomic traits were collected throughout the growth stages of the 176 cannabis accessions. In the vegetative stage, measurements were taken for plant height (H-v), stem diameter (SD-v) at three locations on the main branch using a Fisherbrand digital caliper and measuring tape, leaf count (LC), date of sexual maturity (DSM), number of nodes (NN-v), and canopy diameter (CD-v). During the flowering stage, measurements included plant height (H-f), stem diameter (SD-f), presence of visible axillary stems (AS), number of nodes (NN-f), date of the appearance of the first flowers (dtf), canopy diameter (CD-f), date of trichome appearance (DT), trichome coloration (TC), pistil coloration (PC), and flower maturity date (FMD). For the statistical analysis, plant height at harvest (hh), stem diameter at harvest (sdah), and canopy diameter at harvest (cdh) were used. Additionally, qualitative characteristics related to plant and flower appearance such as leaf shape, similarity between axillary branch and main branch, leaf, trichomes and pistil coloration, maturity data, and disease susceptibility were recorded. In order to assess various growth parameters, growth indexes were generated for plant height (gih), canopy diameter (gicd), stem diameter (gisd), number of nodes (gin), and internode length (giinl). This was achieved by calculating the difference between the data recorded during the first week of flowering and the data obtained at harvest.

Harvesting was conducted when the plants reached approximately 8–10 weeks based on their maturity criteria. This included plants with 60–70% of trichomes displaying a milky coloration and 5–10% of trichomes showing an amber coloration (Appendix A), in addition to other visual elements such as leaf and pistil coloration and the shape and density of flowers (Appendix A). The fresh biomass of the plants was also recorded during harvest. After harvesting, the plants were pruned to keep only the stems with flowers and then dried in the dark for 10 days at a temperature of 16 °C and a relative humidity of 58%. Vertical drying racks were utilized to facilitate proper airflow and drying efficiency. Finally, the plants were manually trimmed using pruning shears, and yield data was collected, corresponding to the weight of the trimmed and dried flowers.

#### 4.5.2. Chemical Analysis

Samples of trimmed and dried flowers, weighing 5 g each (representative of whole plant), were analyzed for cannabinoid content at the Metabolomics Platform in the Institute of Nutrition and Functional Foods (INAF), Université Laval, Québec, QC, Canada, following a modified method described by Mudge et al. (2018). Samples were ground to a mesh size less than 1 mm using a mortar and pestle. A representative sample of 200 mg was then extracted in 25 mL of 80% methanol through sonication for 15 min. The extract was subsequently centrifuged for 5 min at 4500× *g*, and the resulting supernatant was filtered using a 0.22 µm nylon filter and then diluted by factors of 2 and 20. The analysis of cannabinoids was carried out using a UPLC Acquity I-Class system (Waters Corporation, MA, USA) with a PDA UV detector. The compounds were separated on a Cortecs 1.6 µm, 2.1 mm × 150 mm column (Waters Corporation, Milford, MA, USA) maintained at 30 °C. The mobile phase consisted of ammonium formate 20 mM pH 2.92 (A) and acetonitrile 100% (B). The gradient program was set as follows: 0–6.4 min, 76% B; 6.5–8 min, 99% B. From 8.1 to 10 min, the conditions were reinitialized at 76% B. The flow rate was 0.45 mL/min, and the injection volume was 1 µL. Detection was performed at a wavelength of 228 nm. To quantify cannabinoids, 5-point calibration curves were used in the range of 1–100 mg/L for all standards.

To ensure the reliability and consistency of the results obtained from the cannabinoid extraction and analysis method covering 11 distinct cannabinoids, a rigorous validation test was performed. This validation process aimed to assess the accuracy, reproducibility, and overall consistency of the data acquired. The validation test was conducted on a completely random subset of eight accessions (out of 176). Each individual flower sample underwent three separate replicates, involving distinct extraction and analysis procedures. Within this framework, three of the samples were also subjected to triple injections from the same source, allowing for a comprehensive evaluation of the technology’s efficacy (Appendix A). In the validation trial, four cannabinoids, CBN, THCV, CBDV, and D^8^THC, did not meet the minimum detectable quantity threshold and were consequently excluded from our analysis. For these four cannabinoids, only a qualitative assessment was provided, indicating their presence or absence in the samples.

### 4.6. Statistical Analysis

Data preprocessing steps were undertaken, including data cleaning, outlier removal, and handling of missing values. Normality assumptions were assessed using the Shapiro–Wilk normality test based on the distribution of the data [60]. The distribution of data for each trait was computed to determine if it followed a normal distribution. For datasets that exhibited a normal distribution, we conducted analysis of variance (ANOVA), while, for datasets that deviated from a normal distribution, a permutational analysis of variance (PERMANOVA) was performed [61]. To evaluate the relationship between variables, correlation analyses were performed with the PerformanceAnalytics [62] and corrplot packages [63] in R. Regarding the biochemical traits, where the data fell below the minimum detectable threshold, these values were substituted with zeros.

## 5. Conclusions

In conclusion, this study delves into the complexities of cannabis breeding within the Canadian legal market. By closely examining 176 cannabis accessions under controlled conditions, we have gained crucial insights into their agronomic, morphologic, and biochemical traits. Our germination assay and detailed trait analysis have revealed valuable information for precision breeding and cultivar development. We have uncovered how traits related to the maturity time, yield, and biomass are linked, and we have untangled the intricate relationships within physical attributes. Exploring the cannabinoid profiles has unveiled diversity in major cannabinoids, like THC and CBD, and their interactions. A key finding is that the plant’s origin, regular female seeds or feminized seeds, and the choice of using cuttings or seeds significantly affect traits like the maturity time and cannabinoid profile. This research lays the groundwork for future breeding efforts, aiming to develop optimized and genetically stabilized cannabis varieties. With the cannabis industry evolving and contributing greatly to the Canadian and the world’s economy, these insights will shape genetics and breeding practices for the development of medicinal and commercial varieties. These insights will allow for the further exploration and refinement of this plant and shape the future of cannabis cultivation and breeding.

## Figures and Tables

**Figure 1 plants-12-03756-f001:**
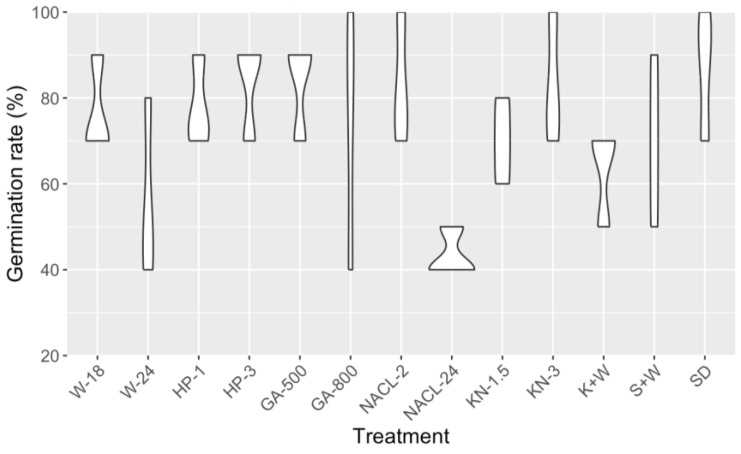
The results of cannabis seed germination test using 13 different treatments. These treatments include water soaking for 18 or 24 h (W-18, W-24), peroxide soaking at concentrations of 1% and 3% (HP-1, HP-3), soaking in gibberellic acid solutions at concentrations of 500 mg/L and 800 mg/L (GA-500, GA-800), as well as soaking in 1% bleach solutions for 2 and 24 h (NaCl-2, NaCl-24), and 1.5% and 3% potassium nitrate solutions (KN-1.5, KN-3). Additionally, two scarification methods were tested: knife scarification followed by water soaking (K + W), and sandpaper scarification followed by water soaking (S + W). A control group with direct sowing (SD) was also included.

**Figure 2 plants-12-03756-f002:**
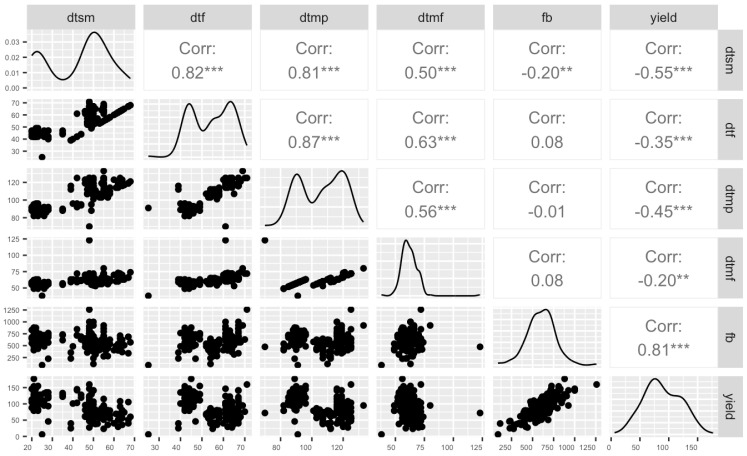
A visual insight into agronomic traits. Scatter plot matrix, distribution, and correlation of coefficients for six agronomical traits: days to sexual maturity (dtsm), days to first flower (dtf), days to maturity from cuttings/sowing (dtmp), days to maturity from flower initiation (dtmf), yield, and fresh biomass (fb). The stars indicate significance levels, ranging from zero to three, where higher star counts signify greater significance.

**Figure 3 plants-12-03756-f003:**
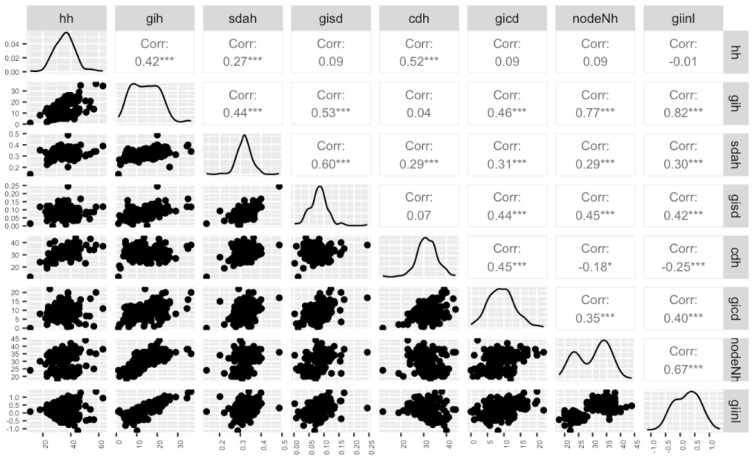
A visual insight into morphological traits. Scatter plot matrix, distribution, and correlation of coefficients for seven morphological traits; plant height at harvest (hh), growth index height (gih), stem diameter at harvest (sdah), growth index stem diameter (gisd), canopy diameter at harvest (cdh), canopy diameter growth index (gicd), number of nodes at harvest (nodeNH), growth index internode length (giinl). The stars indicate significance levels, ranging from zero to three, where higher star counts signify greater significance.

**Figure 4 plants-12-03756-f004:**
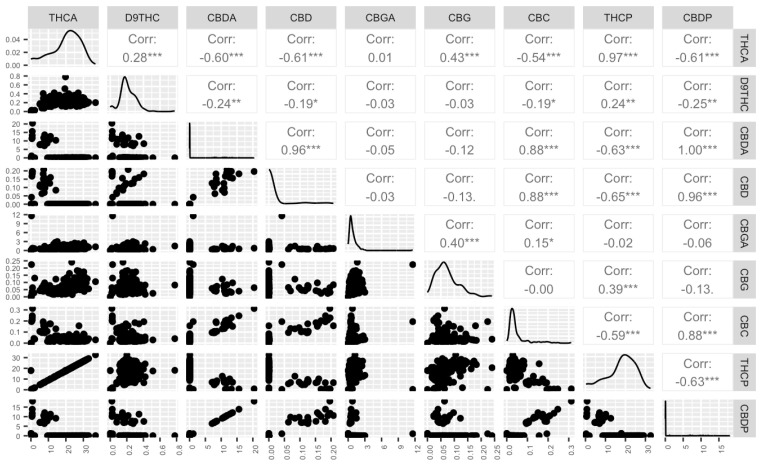
A visual insight into biochemical traits. Scatter plot matrix, distribution, and correlation of coefficients for nine biochemical traits: tetrahydrocannabinolic acid (THCA), delta-9-tetrahydrocannabinol (D^9^THC), cannabidiolic acid (CBDA), cannabidiol (CBD), cannabigerolic acid (CBGA), cannabigerol (CBG), cannabichromene (CBC), potential THC (THCP), and potential CBD (CBDP). The stars indicate significance levels, ranging from zero to three, where higher star counts signify greater significance.

**Figure 5 plants-12-03756-f005:**
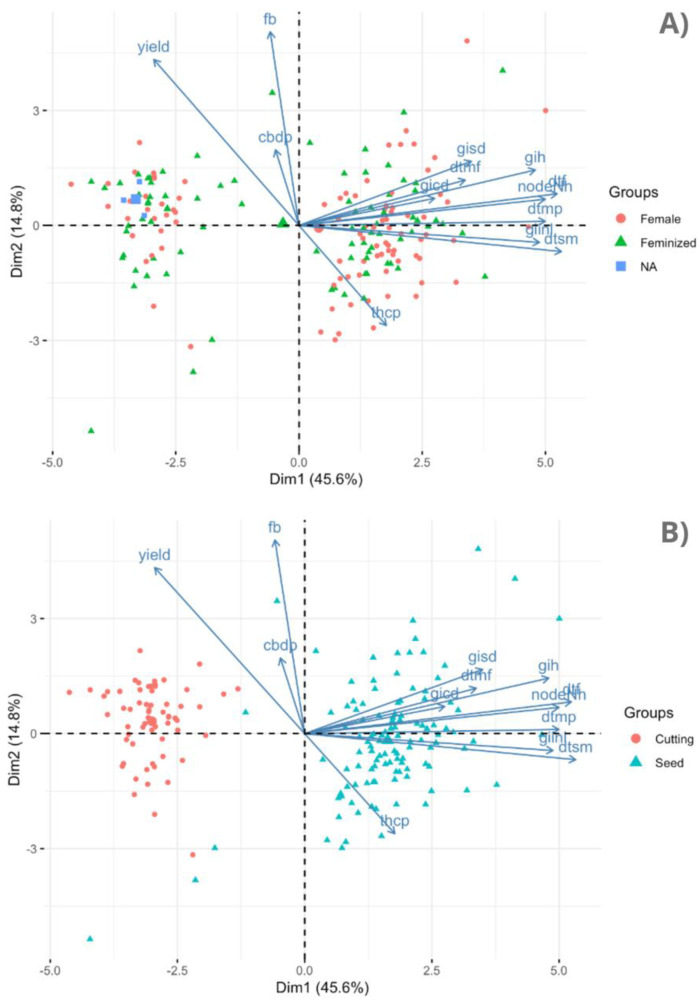
Biplot clustering of cannabis plants based on their origin, regular female seeds vs. feminized seeds (**A**) and their source, cuttings vs. seeds (**B**).

## Data Availability

No new data were created or analyzed in this study. Data sharing is not applicable to this article.

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
