# Peer review of "Comprehensive Phenotypic Characterization of Diverse Drug-Type *Cannabis* Varieties from the Canadian Legal Market"

_plants, 2023, doi:10.3390/plants12213756_

Round 1
Reviewer 1 Report
Comments and Suggestions for Authors
The manuscript entitled “Phenotypic characterization of a diverse population of Cannabis sativa for agronomic, morphological, and biochemical traits” by Lapierre et al. assessed 176 drug-type Cannabis in Canada's legal market, and evaluated various traits (including agronomic, morphological, cannabinoid profiles, etc.). They found that there are negative correlations between the yield and maturity-related traits and positive correlation between the yield and fresh biomass. This study provides insights into the understanding of cannabis’ germination practices, agronomic traits, morphological characteristics and biochemical diversity.
Overall, the method used in the study is thorough. Conclusions are appropriate, and supported by the data. I highly recommend the authors clean the paper before submission.
1. What is the main question addressed by the research?
The authors investigated current drug-type Cannabis in Canada's legal market, and evaluated various traits (including agronomic, morphological, cannabinoid profiles, etc.).
2. Do you consider the topic original or relevant in the field? Does it
address a specific gap in the field?
Yes, the topic is relevant in the field, since it focuses on the drug-type Cannabis in Canada's legal market.
3. What does it add to the subject area compared with other published
material?
The authors presented a comprehensive analysis of Cannabis, but the conclusions are similar to the previous reports (literature 52 and 53). It is necessary to discuss why the authors consider the findings important.
4. What specific improvements should the authors consider regarding the
methodology? What further controls should be considered?
Line 133 : Although the authors describe the methodology in section 4, it is still not clear in detail. i.e. there is no information about Cannabis sativa used for the germination test. How many of Cannabis were selected and tested in the germination test? Similarly, for other tests, the authors should clearly tell the number in the methodology section.
5. Are the conclusions consistent with the evidence and arguments presented
and do they address the main question posed?
"Results section" should focus on describing what authors found when they analyzed their data. But the authors repeated part of the methodology in the results section. They should delete them if not relevant to the data. For example, Line 133 - 137. 156-160, 189-193, 219-232.
The authors should use appropriate subtitles to show the key points instead of the name of methodology. For example,
Line 132: "2.1 Germination test": it is the name of methodology. The authors should consider what kind of key points they found after the germination test.
The author should correct all other subtitles in the results section.
6. Are the references appropriate?
Yes
7. Please include any additional comments on the tables and figures.
Figure 1. There is no significance analysis among 13 methods. It might be helpful to distinguish the best method among them , though with high variances.
Figure 2. Correlation coefficients should be reported to two decimal places. Please correct all of the figures.
Figure 3. The x-axis is not readable due to the formatting. The authors should correct the x-axis.
There are inconsistent names for the same cannabinoids. For example, the author used different short names "potential THC (THCP) and potential CBD (CBDP)" in Line 264, which is different from the previous description " potential THC (THCp) and CBD (CBDp) content in Line 225". The author should keep the name consistent in the study. I found a similar issue for delta-9-Tetrahydrocannabinol in Line 41, Line 222 and Line 238.
Line 221-226 The authors mentioned the quantification of 11 distinct cannabinoids, but they did a validation trial on random eight samples. It is not clear why the authors randomly select 8 out of 11 distinct cannabinoids in the figure 4. Since they are distinct, random 8 samples couldn't be used for assessing the accuracy, reproducibility and overall consistency among 11 cannabinoids. The author should explain the details in the texts.
Supplementary table 2: Line 492-293 The authors mentioned "Overall, a population of 210 cannabis accessions, representing the cannabis drug-type legal market, was assembled for this study (refer to Supplementary Table 2 for details)." But there are only 176 accessions in the supplementary table 2. Please clarify it.
Supplementary figure 5: It seems that the figure legend was incomplete. The authors didn't describe the different panels in the supplementary figure 5. What is the total number of Cannabis sativa L. drug-type varieties used in this study? "175" here. But the authors mentioned "176" in the manuscript.
The reviewer strongly recommends that the authors proofread the whole manuscript, delete comments before submission. And the author should appropriately use the abbreviations in the manuscript. It is not necessary to repeatedly use a full name after showing the abbreviations.
Comments on the Quality of English Language
Proofread the manuscript and remove the comments
Reviewer 2 Report
Comments and Suggestions for Authors
Overall it is a worthy study. Some comments:
1. l.117 'invaluable' means not valuable. I know that you see it used often but proper English uses 'valuable'.
2. l133-137 should be in Materials and Methods section along with l. 156-160, 189-193, 219-232.
3. Figure 2. significant digits should be limited to 2.
4. Germination test was only conducted on one cultivar???
5. l.471- what is the ideotype plant ?
6. l347-350 correlation already reported. as well as l.350-352.
7. l. 407-409 tradeoff between yield and THC already reported.
8. l. 445-449 so isolines were not used to determine cuttings had higher CBD and THC ???
9. Section 4.5 how many plants were measured for morphological traits >
10. l. 646-648 so key findings have already been reported ??? What is new then?
Round 2
Reviewer 2 Report
Comments and Suggestions for Authors
The misused word 'invaluable' is found again in line 669.
L. 140 cannot say 'highest' when not significant. You can say ' numerically highest'
I still don't understand the 'ideotype' wanted.
I still don't think you can definitively say 'cuttings have higher CBD and THC than seed -derived plants when you did not use isolines.
Need titles to supplementary tables and figures. It helps the reader.
Supplementary Table 2. No need for decimal. It is strange that all the numbers were whole to 2 places !
Author Response
We thank again the reviewer’s efforts and constructive comments on this manuscript. All comments and suggestions have been carefully addressed in the revised manuscript.
Responses:
The misused word 'invaluable' is found again in line 669.
Corrected.
140 cannot say 'highest' when not significant. You can say ' numerically highest'
Corrected as recommended.
I still don't understand the 'ideotype' wanted.
We add more text to describe the ideotype of the plant.
"The germination tests were performed using seeds from an inbred hemp-typed cannabis cv. Vega. Vega is a grain hemp cultivar characterized by its short plant stature, exceptional lodging tolerance, uniform height, and flowering, and it typically exhibits a very low male count. The seeds from this cultivar contain approximately 0.2% THC and 1.6% CBD."
I still don't think you can definitively say 'cuttings have higher CBD and THC than seed -derived plants when you did not use isolines.
We acknowledge the concerns raised by the reviewer, and as a response, we have included additional text in the Discussion section to provide further clarification.
"Nevertheless, it's crucial to emphasize that additional research is essential to validate these findings, ideally through the use of isolines for more conclusive results."
Need titles to supplementary tables and figures. It helps the reader.
Titles and legends were added.
Supplementary Table 2. No need for decimal. It is strange that all the numbers were whole to 2 places !
Corrected.